# Comparative Analysis of Ultrasonography and MicroCT Imaging for Organ Size Evaluation in Mice

**DOI:** 10.3390/jimaging11060200

**Published:** 2025-06-18

**Authors:** Juan Jose Jimenez Catalan, Marina Ferrer Clotas, Juan Antonio Camara Serrano

**Affiliations:** 1Preclinical Imaging Platform, Vall d’Hebron Institute of Oncology, 08035 Barcelona, Spain; juanjosejimenez@vhio.net; 2Gnotobiotic Core and Mouse Metabolism Core, University of California San Francisco, San Francisco, CA 94143, USA; marina.ferrerclotas@ucsf.edu; 3Preclinical Therapeutics Core, University of California San Francisco, San Francisco, CA 94158, USA

**Keywords:** microCT, ultrasound, preclinical imaging, organ volume, mice

## Abstract

In this work, the authors compared microCT and in vivo ultrasonography in terms of accuracy and efficacy for measuring the volume of various organs in mice. Two quantification protocols were applied: ellipsoidal volume measuring maximum diameters in all three axes in both imaging systems and manual delineation of organ borders in microCT studies. The results were compared with ex vivo volumes. In general, both imaging techniques and quantification protocols are accurate, but ultrasound is faster in both acquisition and analysis. The only accurate method for heart volume measurement is manual segmentation on microCT. For the ovary, none of the techniques and protocols had a positive correlation with ex vivo volume. The three-diameter method can be used for ellipsoid organs because of its rapidity, but for more irregular structures, manual segmentation is recommended, although it is time-consuming.

## 1. Introduction

In recent decades, animals have been used as human models in medical research. For basic research, mice and rats are the most commonly used species, accounting for up to 85% of the animals used in research [1]. Downsizing (reduction) is an essential concept in biomedical research and should be applied in all experiments involving live animals along with refinement and replacement for ethical considerations [2]. Still, after reducing the sample size of experimental groups to the statistically minimal size, experiments with these species often involve large groups of animals, which entail a long time for each animal-related task. To improve these processes, efficient protocols and routines during in vivo experiments are desirable.

Preclinical imaging aids animal research by providing valuable information on animals without the need to euthanize them. These techniques allow repeated examinations on the same subjects, reducing the number of animals used and facilitating the comparison of the same animal with itself, improving statistical power. As for the available imaging techniques, most are a direct translation from human medicine, with the necessary engineering modifications to obtain valuable images and information from small subjects, like higher spatial and temporal resolutions or faster acquisition times. The most relevant technologies are micro-computed tomography (microCT), positron emission microtomography (microPET), preclinical ultrasound, or preclinical magnetic resonance imaging (MRI). The only technology originally developed for laboratory animals is Optical Imaging, which is now beginning to be explored in human medicine [3,4,5,6]. All these techniques can be classified into two main groups, depending on the nature of the information generated: anatomical/physiological and metabolic/molecular. The first group includes techniques that can report on the anatomy or physiology of the target, such as organ size, blood velocity, or the motion of a structure. We can include in this group microCT, ultrasound, and MRI, which inform us about different anatomical and physiological parameters depending on technology. In the second group are microPET, microSPECT, optical imaging, and variations in ultrasound and MRI (targeted contrast agents in ultrasound and BOLD and spectroscopy in MRI). With these imaging techniques, we can obtain information on the metabolism or molecular activity of structures [7,8].

One of the main uses of anatomical techniques is the measurement of the size of structures, comparable to what is performed in humans for the evaluation of diseases in different areas such as cancer or metabolic pathologies. For example, the evaluation of tumor size in oncology. In summary, after the scan, the image analysis starts with a visual examination of a specific anatomical region in search of the target structure. Once the structure is found, its volume is measured. At this point, the standardization of measuring protocols is key to avoid errors or fluctuations in the results.

There are different approaches to measuring the size of a structure in imaging studies, both in clinical and preclinical work. One of the simplest is the so-called “geometric approach”. This method assumes that the structure has a regular geometric shape and therefore different diameters are measured, and a mathematical formula is applied to obtain the volume of the structure. It is fast but is mostly limited to structures of regular geometric shape. Even so, it is widely used in clinical medicine because of its ease and speed. Another method is the manual segmentation of the structure. It requires a three-dimensional study, such as those that can be obtained from microCT, microPET, MRI, or ultrasound with a motorized scanning platform. The examiner delimits the perimeter of the structure in each slice that composes the structure, and at the end a three-dimensional object is obtained and its volume is calculated. This method is more time-consuming, but can be applied to any structure without limitations related to shape or morphology. And it can only be used in three-dimensional studies. Different variations and attempts to automate this protocol have been developed, including the Seeds and Frontiers method, threshold-based segmentation, and other similar methods [9,10,11,12]. In all these methods, the examiner needs to draw some perimeters of the structure, and the software will automatically interpolate and segment the structure and calculate the volume. In recent years, new methodologies have been developed, taking advantage of the progress of imaging software. The use of anatomical atlases for organ segmentation is a recent approach in preclinical imaging. The idea of this approach is the use of pre-existing maps of animal anatomy, with the different organs already segmented, that somehow can be overlapped with the study. The software will be able to segment the current dataset using the segmentation loaded from the atlas as a reference, adjusting the size and orientation of both studies [13].

Abdominal and thoracic organs frequently change their volume during the course of pathology, often as paraphysiological compensation for organ failure, or due to being directly affected by a pathological process. Numerous diseases involve an alteration of the organs that respond to the imbalance of homeostasis by increasing or decreasing their volume due to blood stasis or dehydration. Examples of this volume change are cardiomegaly following compensated heart failure; splenomegaly in infectious diseases, immune pathologies or blood cancer; renomegaly in unilateral renal failure and hypertension; testicular enlargement due to orchitis or tumors; and ovarian enlargement due to ovarian cyst or tumor growth. During the course of pathology, monitoring the size of the affected organs is useful to assess the progression of the pathology or the effect of treatments. For these purposes, it would be advantageous to use an in vivo imaging technique with standardized protocols that could be applied on a large scale.

The current scenario of large sets of animals and different calculation protocols led our group to elucidate what is the best approach for organ volume calculation. We designed an experiment using the ultrasound and microCT systems accessible in our animal facilities, but a similar approach could be applied to other anatomical techniques such as MRI.

## 2. Materials and Methods

Twenty-three mice (seventeen females and six males) with multiple genetic backgrounds were included in this study, including C57BL/6, Athymic Nude-Fox N1, NOD.Cg-Prkdcscid Il2rgtm1Wjl/SzJ and NOD.Cg-Rag1tm1Mom Il2rgtm1Wjl/SzJ mice. The inclusion of immunocompetent and immunodeficient animals ensured a variety of backgrounds in different organs, especially immune-related structures. All of these mice were naïve and came from different and unrelated experiments. The mean age was 169 days (45–255 days). The mean weight was 25.71 g (18–39 g). The mixture of strains, sexes, ages and weights was ideal for obtaining inhomogeneous organ volumes. An excessive homogeneity of structures could be a confounding factor in this project. The biological data of the sample are given in the Appendix A: Biological data of the sample.

The list of organs evaluated during this project includes the heart, left and right kidneys, spleen, left and right ovaries, left and right testicles, and left and right inguinal lymph nodes.

Ultrasound scans were performed with a Vevo2100 system, using Visualsonics-Fujifilm (3080 Yonge Street, Suite 6100, Toronto, ON, Canada). The scanning frequency was adjusted to the target size, in the range of 20 to 30 Mhz. The frame rate was set at 20 frames per second and the gain in the range of 22 to 39 decibels. The scanning depth was 1 cm.

The microCT system used during the examinations was a Quantum Gx2 from Revvity (940 Winter St, Waltham, MA, USA). The scanning field of view (FOV) was set at 36 mm and the reconstruction FOV at 25 mm, with an acquisition time of 14 s for the entire scan. X-ray energy was 70 kilovolts and 114 microamperes. An X-ray filter “0.06 mm Cu + 0.5 mm Al” was applied to reduce noise in the images. The spatial resolution of the reconstructed images was 50 × 50 × 50 µm voxel size.

All studies were acquired and analyzed by an operator with 15 years of experience (JCS). Animal procedures followed the guidelines of the IACUC of the University of California, San Francisco. The protocol number was AN194778.

For image acquisition, mice were anesthetized with a mixture of isoflurane (Isoflo^®^, Zoetis, 10 Sylvan Way, Parsippany, NJ, USA) in fresh oxygen at a flow rate of 1 L per minute. The concentration of isoflurane varied from 5% during induction of anesthesia to 2.5% for the maintenance phase. The mouse was placed on the prewarmed ultrasound platform in dorsal recumbency for ultrasonography. Thoracic and abdominal skin was trimmed (Oster Vorteq, Oster, 2381 NW Executive Center dr, Boca Raton, FL, USA) and shaved with depilatory cream (Nair. Church and Dwight co., Ewing Township, NJ, USA). Ultrasound gel (Aquasonic, Parker laboratories, 4 Sperry Road, Fairfield, NJ, USA) was applied to the skin. Target organs were scanned in B-mode in axial and longitudinal views. The widest diameters were obtained and recorded for each plane.

Once the ultrasound examination was completed, the animal was transferred to the microCT system. The anesthesia was maintained with the same oxygen flow and isoflurane rate. An iodinated radiological contrast agent (Omnipaque^®^, GE Healthcare, 1283 Mountain View-Alviso Rd, Sunnyvale, CA, USA) was injected intravenously at a dose of 3.5 mg iodine per gram of body weight, and a thoracic scan was first acquired to obtain images of the enhanced heart. Next, the caudal part of the abdomen was scanned, focusing on the testicles and inguinal lymph nodes. A second dose of the same contrast agent and at the same dose was performed, this time intraperitoneally. This last injection helped to delimit the spleen, ovaries, and kidneys. The last scan focused on the cranial and mid-abdomen region. In total, three microCT scans were obtained in each animal. The image acquisition process kept the mice anesthetized for 25 to 30 min.

After finishing the microCT imaging, the anesthetized mouse was euthanized with CO_2_, followed by cervical dislocation, and the target organs were removed and weighed on a precision balance (Ohaus, Ohaus Corporation, 8 Campus Drive, Suite 105, Parsippany, NJ, USA).

Ex vivo volumes were calculated by applying the mathematical formula for density, using organ-specific density values. These values were obtained from previously published work [14] and are displayed in the Appendix A: reference tissue densities. The density formula states:Density = weight/volume

In our case:Volume = weight/density

The resulting ex vivo volumes were considered as gold-standard values for evaluating the accuracy of imaging-based volumes.

For the analysis of ultrasound images, the organ volume was calculated including the measured diameters in the ellipsoidal volume formula:Volume = 4/3π × a × b × c
where “a,” “b”, and “c” are the radius values for each plane.

Two forms of analysis were used for microCT images. First, the largest diameter per plane was measured, similar to the ultrasound analysis, using a dedicated software from the microCT (Quantum Gx control software, version 6.3.2 Rigaku© 3-9-12, Matsubaracho, Akishimashi, Tokyo, Japan). For the second analysis, the manual segmentation of the microCT images, the studies were transformed into DICOM files and analyzed using Slicer software, version 5.8.1 [15]. Three-dimensional volumes were obtained by manually delimiting the edges of the structure. The organ volume was obtained automatically once the organ segmentation was completed.

Figure 1 and Figure 2 show examples of ultrasound and microCT diameter measurements. Figure 3 shows examples of manual segmentation.

To check the efficiency of the analysis, the time required for each image analysis was recorded. The statistical analysis consisted first of a description of the dataset, including mean, median, range, and standard deviation values. Next, correlation analyses were run on each organ by comparing the gold-standard value, the ex vivo sample volume, obtained by dividing the organ weight by the organ density; and the other volumes, namely ultrasound imaging volume, measuring the three diameters for each organ in the ultrasound images; microCT diameter volume, measuring the three diameters in the microCT images; and microCT manual segmentation volume, segmenting the organ manually from microCT datasets. As for accuracy assessment, the ex vivo sample volume was used as the gold standard, and the remaining volumes were compared with it to elucidate the most accurate way to obtain the organ volume using an in vivo imaging technique. As a complementary analysis, the ex vivo sample volume of each organ was compared with its contralateral in bilateral organs. The presence of significant discrepancies between contralateral organs could be indicative of an error in organ harvesting, assuming that contralateral structures should have similar volumes in healthy animals. An error in the sampling procedure could affect the veracity of the results, knowing that this is considered the gold-standard value. All statistical analyses were performed with Sigmaplot^®^ software version 15.0 (Graffiti LLC 405 Waverley St Palo Alto, CA, USA).

## 3. Results

A total of 184 samples were analyzed during the experiment, derived from 23 animals across 8 target structures. Each sample was analyzed with three imaging analysis protocols. As a result, the total number of data points was 552.

The mean volume values for each organ and imaging protocol plus sample volume are listed in Table 1.

Statistical results are shown in the Appendix A: Statistical results of organ volumes, Appendix A: Time required for each organ and imaging technique, and Appendix A: Correlation analysis of organ volumes and different imaging techniques. The diagrams of the different results are shown in Figure 4.

As for the time required, image acquisition by ultrasonography ranged from 10 to 15 min. In the case of microCT, including contrast agent injection, the acquisition time was similar, in the order of 10 to 15 min per animal. During image analysis, measurement of the different diameters in the ultrasound images required one to two minutes per organ, regardless of the target. For microCT, the measurement of the diameters required more time, around 3–4 min per organ, due to the need to reorient each structure in each plane for measurement. For manual segmentation analysis, the time required varied from 5 min for small structures, such as lymph nodes or ovaries, to 15 min for larger structures, such as the heart or testicles. In summary, ultrasound measurements of all organs in a single mouse required 15 to 20 min, in contrast to microCT, which required 25 to 30 min for diameter measurements or 45 to 60 min for manual segmentation. A summary of these results is shown in the Appendix A: required time for each organ and imaging technique, and Figure 5.

Regarding the accuracy of the different techniques and protocols, the results of the correlation analysis are shown in the Appendix A: Correlation analysis of organ volumes and different imaging techniques and Table 2. Compared to the reference value (ex vivo sample volume), a positive correlation of this volume and the others (ultrasound volume, microCT diameter volume, and microCT manual segmentation volume) was observed in most of the measured organs, including lymph nodes, spleen, kidneys, and testicles. In all these organs, the correlation was strong between ultrasound, microCT (both quantification methods) and ex vivo sample volumes. Between ex vivo volumes in contralateral structures (lymph nodes, kidneys, and testicles), the correlation was also positive, confirming the correct sampling methodology.

Regarding the heart, the manually segmented volume by microCT presented a positive correlation with the ex vivo sample volume (*p* < 0.05) but not the other methods based on ultrasound or microCT diameter measurements (*p* > 0.05).

For the ovaries, there was a correlation between ex vivo sample volume in contralateral structures, but no correlation was found between the three imaging techniques and ex vivo sample volumes. As a result, no technique or protocol was accurate for the calculation of ovarian volume.

## 4. Discussion

The aim of our study was to elucidate the efficiency and accuracy of measuring volumes in different organs in mice, using two imaging techniques frequently available in preclinical research (ultrasonography and microCT) and three quantification methods (one for ultrasonography and two for microCT). In addition to the accuracy of the results, we inquired about the efficiency of the process, so we recorded the time required to scan and quantify each organ. In in vivo research, the study cohort usually includes a considerable number of animals, so the time for each individual analysis must be multiplied by the whole experiment. In that case, differences in a few minutes between techniques can mean hours in real experiments. To our knowledge, this is the first attempt to compare the accuracy and efficiency of these techniques for organ volume calculation. After extensive research in both clinical and preclinical publications, the general tendency for organ volume measurement is to follow previously published papers and protocols, without ever asking whether another approach might be possible. Studies in which the authors compare their results with a reference volume, such as the ex vivo sample weight in our case, are rarer. In general, organ volumes are compared between experimental groups in the same project.

The difference between ultrasound and microCT in terms of data acquisition and analysis is obvious. In the case of ultrasound, the entire scan requires 10–15 min plus a similar time for image analysis, resulting in 25–30 min for the entire process. The same procedure using microCT needs a similar time for image acquisition, but depending on the analysis protocol, the time differs significantly. Finding the largest diameters in all three planes requires about 40 min for all organs due to the difficulty of rotating the studies. This process is faster in ultrasound by moving the scanning probe manually. Manual microCT segmentation requires more time for analysis due to the need to segment each organ slice by slice. Depending on the size of the organ, it can take from 5 min for ovaries or lymph nodes to 10–15 min for the heart or kidneys. In addition, in microCT studies, intra-abdominal organs may be difficult to find and analyze, even with the aid of contrast enhancement. The kidneys and spleen may be easy to find, but the ovaries may be more difficult due to the lack of soft tissue contrast between nearby structures such as intra-abdominal lymph nodes, uterus, infundibulum, or empty bowel segments.

The main disadvantage of ultrasonography is its dependence on the operator, which requires prior knowledge and experience to obtain reliable results [16,17,18]. In the case of microCT, data acquisition is almost automatic, but image analysis can be difficult and also operator-dependent. In our study, all images from both systems were acquired and analyzed by the same experienced operator, so individual bias could be considered neutralized, although conclusions about the time required may be affected by this experience and previous knowledge.

Based on the results, manual microCT segmentation is the most accurate analysis protocol. The results of this analysis match the sample volume in all organs except the ovaries. In the case of the heart, it is the only approach accurate enough to measure its volume. In our opinion, this is because the other quantification methods are based on the ellipsoidal volume formula, but the heart is not an ellipsoidal-shaped structure, so the results of this approach do not match the actual volume of the heart.

For the analysis of the heart, this study considers the organ as a whole and cardiac measurements were performed on the entire structure, without dividing the different cardiac chambers as is frequently performed in cardiology. In addition, the heart beats and moves during the cardiac cycle and these changes in shape and size are difficult to compensate for when performing microCT imaging, unless a gated study is performed. In ultrasonography, due to its dynamic nature, it is easier to record the entire cardiac cycle and select, during analysis, which of the image frames is best suited to obtain the largest diameters. In our case, we performed a static microCT without cardiac gating, and as a result, the complete study is composed of several images of different cardiac cycles. Putting them all together, the software creates a static image of the heart. As for the ultrasound, we recorded video clips of several cardiac cycles, and the maximum diameters were guaranteed thanks to this dynamic view. As for harvesting, the sampled heart retained some of the blood contained when the organ was weighed, so it was included as part of the cardiac volume when the ex vivo volume was obtained. The difference in densities between blood and cardiac tissues is, in the author’s opinion, minimum (heart: 1.053 mg/mm^3^, blood: 1.060 mg/mm^3^), and assuming blood as cardiac tissue would not affect the volume calculation in a significant way. Furthermore, it will require some time to remove the blood from the heart chambers before weighing the organ and even more time to do the same during the manual segmentation of the heart in the image analysis. For the diameters approaches in both ultrasound and microCT, it would be impossible to compensate for the presence of this blood during the volume calculations. In parallel, the blood was included as part of the cardiac volume when analyzing the images. In summary, in the different cardiac volumes calculated, there was blood counted as cardiac volume regardless of how the volume was measured.

In comparison with other authors, most publications related to cardiac volume focus on left ventricular (LV) volume to obtain cardiological parameters such as systolic volume or cardiac output. Different imaging techniques, such as microCT [19,20,21], are used to obtain this LV volume. In these publications, thresholded segmentation and manual segmentation after injection of a contrast agent were used, following an approach similar to ours. Badea et al. compared different radiological contrast agents to clarify which one is the best for segmenting the LV and calculating different cardiac parameters [19]. They used threshold-based segmentation to measure the heart size, but did not compare their results with any biological data, only looking for differences in left ventricular volumes between the different contrast agents. Befera et al. compared volumes between three different imaging modalities (single-photon emission-computed tomography (SPECT), microCT, and MRI), but did not compare the different results with any biological specimen but with MRI, which was used as the gold standard [20]. Holbrook et al. defined two novel protocols for gated microCT reconstructions and segmented the LV with a “Seeds and Borders” protocol (based on thresholded segmentation) [21]. Again, no comparison with ex vivo samples was presented. All these studies described gated cardiac and respiratory imaging to obtain detailed images of the left ventricle. In our opinion, it is clear that there is a change in heart volume between the systolic and diastolic phases, and if working in detail with the heart, gated microCT images are desirable and optimal. In our work, we required a quick and easy way to measure cardiac volume, and gated images require more time for image acquisition. Our results confirm that manual segmentation of static studies could be sufficient to obtain a volume close enough to ex vivo.

Regarding ultrasonography, Collins et al. explored the possibility of using three different approaches for LV volume calculation by B- and M-mode ultrasonography, including the ellipsoid and truncated cone approaches, in a manner equivalent to our work [22]. They concluded that all methods overestimated necropsy weights as we did, and that the ellipsoid B-mode approach was more accurate than the M-mode calculation. Unlike our work, they did not have any three-dimensional technique to work with, so no three-dimensional segmentation of the heart was performed in their study.

The other structure that, surprisingly in our opinion, does not match the imaging volumes are the ovaries. These structures have a round shape, so theoretically, the ellipsoidal approach should be effective; but the reality is that none of the imaging techniques or protocols were sufficiently accurate compared to the gold standard volume. Only the ex vivo sample volumes in contralateral structures were statistically similar, confirming the correct sampling method used during the project. In the imaging part of the process, the ovaries are easily scanned with ultrasonography and are depicted as hypoechoic structures caudal and lateral to the caudal pole of the kidney. They were found in all females and measurements were possible. In the case of microCT, the localization of the ovaries can be difficult due to the lack of contrast between neighboring structures, even after intraperitoneally injecting a radiological contrast agent. In addition, the ovary is intricately connected to the infundibulum, and it may be difficult to separate the two structures during organ delineation in imaging and weighing. This could explain the lack of correlation between ex vivo volume and imaging volumes, including ultrasound.

In future studies, the authors are planning different approaches to imaging the ovary depending on the technique, trying to resolve measurement errors. By injecting a non-radiological solution such as PBS or saline prior to ultrasonography, the entire ovary could be surrounded by an anechoic solution, which would create sufficient contrast between the soft tissue structures. Thus, our hypothesis is that it would be possible to find the actual boundaries of the ovary. For microCT imaging, the approach would use a radiological contrast agent as we did in the current project, but a diluted one, in order to reduce artifacts in the images. Our plan is to try different dilutions of contrast and find the best concentration to segment the ovary. In our opinion, the small size of the ovaries does not justify the discrepancy between volumes. The lymph nodes in immunodeficient mice were smaller and their results matched between imaging and ex vivo results. Another point to keep in mind when studying the ovaries is that these structures are under the effect of sex hormones and change their volume cyclically [23] due to the growth of ovarian follicles and the release of eggs. This fact was one of the reasons why the authors decided to perform the whole experiment uninterrupted, without intervals between image acquisitions or organ sampling. In our opinion, this is the only way to minimize this variability of ovarian size.

Reviewing published works, different imaging techniques have been used for ovarian analysis, but ultrasonography is the leader in this contest due to its better soft tissue contrast compared to X-ray-based technologies. Amaral et al. manually segmented the ovary using optical coherence tomography (OCT), comparable to our approach with microCT [24]. Their technology had a higher spatial resolution compared to microCT, so they compared their results with histological images rather than macroscopic ex vivo specimens. Therefore, the correlation between OCT and histology was positive. Mircea et al. used ultrasonography to measure the ovary and its follicles [25]. They measured the volume of the organ using the widest diameters and the ellipsoid formula, in the same way that we did. Unfortunately, they did not weigh the organ after sampling, but directly processed the tissues for histology. In a different field, ultrasonography is often used to check the ovarian cycle in farm animals, such as cattle, for reproductive reasons. Jaiswal et al. measured the ovary and follicles in the same way as we did, using the wider diameters to calculate volume [23]. Again, they did not compare their results with ex vivo data.

As for the rest of the organs, all of them have an ellipsoidal shape, which makes the volume calculation precise enough to obtain volumes that match the gold-standard volume of the organ. Examples of this fact are the kidneys and the testicles. In both cases, segmentation and volume measurements were feasible.

In the case of kidneys, the ultrasound scan was performed easily thanks to the good soft tissue contrast that this technique has. For the microCT imaging, the intravenous injection of radiological contrast enhanced the organ sufficiently to easily segment the structure. There are several publications that use different in vivo imaging techniques for renal volume calculations. Baldelomar et al. analyzed the renal size with MRI and used a manual segmentation of the organ plus a mathematical iteration protocol [26]. Unfortunately, they only used ex vivo samples to compare renal nephrons, but not the whole kidney. Wallace et al. calculated the renal volume by drawing the perimeter of the organ in different slices of coronal or axial views on MRI images [27]. They compared the imaging results with ex vivo volumes in the same way as we did, calculating volume as a function of sample weight, although they assumed that renal density was like that of water, whereas we tried to do it in a more precise way and used a specific renal density obtained from the literature [14]. Müller and Meier presented a compilation of renal volume measurements on MRI images and defined three different ways of volume calculation: ellipsoidal volume approach, voxel counting method, and median cut-off technique [28]. The first two coincide with our work, and the third is a modification of the ellipsoidal protocol. In detail, they measured the renal area in the center of the organ and multiplied it by the number of slices, adjusting the results with a coefficient. They concluded that the voxel counting method is more accurate, although more time-consuming, analogously to our results.

In clinical medicine, there are numerous papers on renal volume calculation, with most publications concluding that the ellipsoidal approach is the best way to rapidly calculate renal volume. Examples of these papers are Park et al. [29], Jones et al. [30], and Breau et al. [31]. In all these studies, there was a recurring sentence: “Comparison of the results revealed that renal volume can be determined by the simplest method, the ellipsoid method, with sufficient accuracy for clinical use”.

Since mouse testicles are retractile, they were first moved into the scrotum by applying light manual pressure on the caudal part of the abdomen. Looking at the literature, little work has been performed recently on testicular volume in mice. Most authors extracted the testicles and weighed them to analyze their size. As for imaging, Yokota et al. measured the testicular volume by MRI and drew regions of interest on each slice, as we did with our microCT images [32]. They looked for a potential testicular toxicity of busulfan and only compared the treatment groups, but no comparison with ex vivo samples was performed in their work. Spears et al. compared the testicular size between manual measurement and ultrasonography [33]. For ultrasonography, they used the ellipsoid hypothesis for volume calculation, but interestingly, they were unable to measure the length of the organ due to difficulties in differentiating the testicular pole and the epididymal head. Finally, they compared both techniques with ex vivo volumes. Their results differed from ours in terms of accuracy; they did not find a correlation between ultrasound and ex vivo volume, whereas we did. Apart from the potential operator influence on the results, these authors used a clinical device for imaging, with a probe not adapted to laboratory animals, so part of the inconsistencies and difficulties they encountered could be related to technical limitations. Interestingly, they measured the time required for the ultrasound examinations of the testicles and their results agree with ours. The examination required 2–3 min per organ, as in our case, but they did not measure testicular length, so theoretically, their exams would take longer if all the diameters were measured.

Lymph nodes were another of our target organs. These structures can change in size in different situations, such as infections, cancer or tissue inflammation. We measured the inguinal lymph nodes, which are easily accessible, compared to other lymphatic structures, such as, for example, mesenteric or periportal lymph nodes. The inguinal lymph nodes are surrounded by adipose tissue, and this enhances the contrast between the node and the surrounding tissue. This occurs on both ultrasound and microCT images, even without injecting contrast agents. We were able to find the lymph nodes even in immunosuppressed animals, whose size is smaller. In our results, the volumes obtained with both imaging techniques have a good correlation with the ex vivo volumes. In comparison to published work, Vitiello et al. measured regional lymph nodes in a melanoma model with ultrasonography and their approach for volume quantification was a multi-slice contouring method in 3D ultrasound studies [34]. They used the contralateral lymph node as a control and did not compare the imaging volumes with any ex vivo data. Similarly, Walk et al. performed a quantification of the cervical lymph node using slice contouring with three-dimensional ultrasonography [35]. They presented an examination protocol, but no reference value was available for comparison. In a different direction, Yamaki et al. used the ellipsoidal approach to measure axillary lymph nodes with ultrasound and microCT techniques, but the microCT images were performed ex vivo and for microvascularization analysis, so no nodal volume was measured and no comparison between techniques was performed [36]. Finally, Proulx et al. used a threshold-based segmentation to measure the volume of synovial lymph nodes in an arthritis model [37]. As reference, they used histological slices with maximal cross-sectional area to calculate the node volume. They demonstrated a good correlation between imaging and histological results.

Finally, the spleen was easy to find on both ultrasound and microCT after injecting the radiological contrast agent intraperitoneally. The organ is located craniolaterally to the left kidney and has a slight ellipsoid shape, which helps to obtain accurate measurements. In our work, we were able to locate and quantify the volume of the spleen in both imaging techniques and all results correlate with ex vivo volumes. Looking at the literature, the spleen has often been monitored as an indicator of immune responses and other blood malignancies. Splenomegaly is a common finding in pathologies in different fields of medicine, from immunology to cancer and infectious diseases. Therefore, different attempts have been proposed to measure the spleen volume with in vivo imaging techniques. For example, Hodgson et al. used ultrasonography to check spleen size in colorectal cancer [38]. They measured all three axes in their major and minor diameters and used a variation in the ellipsoidal approach for the final spleen volume. Paterson et al. imaged the spleen with MRI and used manual segmentation at each slice as a protocol [39]. They applied the density value of 1.089 g/cm^3^ to calculate the mass of the spleen. This value is higher than that used in our work (1.05 mg/mm^3^). Similarly to Paterson, Jackson et al. segmented the spleen with manual contouring in individual 2D slices and propagated the contours through the rest of the slices to measure the volume by MRI [40]. They corrected the organ volume for whole body mass and compared the results between different treatment groups, without ex vivo data.

Summarizing the results, we can say that ultrasound is accurate enough to be applied to organ volume calculation when using an ellipsoidal volume protocol. This technique is faster than microCT and no radiation is applied to the subject. This fact could be relevant for in vivo experiments in which animals are to be scanned repeatedly. Radiation can affect tissue biology, and the use of ultrasound would avoid this. The main weakness of ultrasonography is its dependence on operators. This can be somewhat avoided with the recent implementation of motorized scans, which obtain three-dimensional studies. With this application, an entire structure can be imaged, and the analysis can be performed off-site, even by a different operator. Data acquisition becomes partially automatic, similar to microCT imaging.

The manual segmentation of microCT studies is the most accurate method and can be applied even to irregular structures. Moreover, the technique is less operator-dependent than ultrasonography, but not completely independent, since the perimeter tracing of organs is performed manually. The weakest part of this method lies in the fact that it requires more time for analysis, and, in addition, the samples are exposed to radiation beams. As mentioned above, radiation can have an effect on biological processes, and this effect increases with the radiation dose and the number of exposures.

This project had several limitations. First, the microCT analysis of the heart was performed without cardiac gating, so only a static volume of the organ could be obtained. Second, both techniques are somewhat operator-dependent. We tried to avoid biases in the results with a single operator. Still, the results in terms of time required for acquisition and analysis should be taken with caution due to this fact. The number of samples may be another limitation, especially the difference between females and males, although the aim of this project is not a comparison between sexes or sex organs. The sample size for each organ was large enough to obtain statistical significance. Finally, the lack of a motorized ultrasound system made it impossible to obtain three-dimensional studies, so that, in the case of ultrasonography, only one method of analysis could be performed.

## 5. Conclusions

The calculation of organ volumes by in vivo imaging techniques is feasible and accurate compared to ex vivo volumes, except in the case of the ovaries, which were underestimated in both equipment and analysis methods. In the case of the heart, only manual segmentation with microCT is sufficiently accurate to obtain organ volumes.

In terms of process efficiency, ultrasound image acquisition is similar to microCT. The main difference in time is based on the analysis protocol. The ellipsoid-based volume protocol is faster, but fails in the analysis of the heart. Manual segmentation is more accurate but requires more time.

In our opinion, both microCT and ultrasound are acceptable imaging techniques for organ volume calculation, and the choice between them depends on the target organ, equipment availability, and operator experience.

## Figures and Tables

**Figure 1 jimaging-11-00200-f001:**
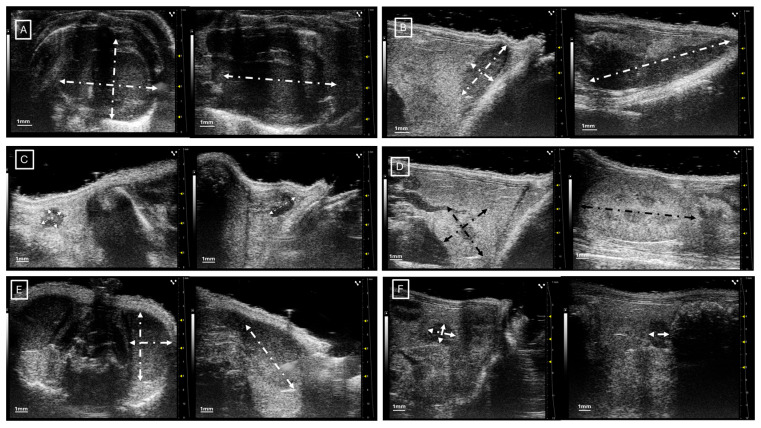
Representative images of ultrasound diameter measurements in (**A**) heart in axial (left) and longitudinal (right) views, (**B**) spleen in axial (left) and longitudinal (right) views, (**C**) inguinal lymph nodes in axial (left) and longitudinal (right) views, (**D**) left kidney in axial (left) and longitudinal (right) views, (**E**) left testicle in axial (left) and longitudinal (right) views, and (**F**) left ovary in axial (left) and longitudinal (right) views. In all cases, the diameters are marked with dotted, white, or black lines.

**Figure 2 jimaging-11-00200-f002:**
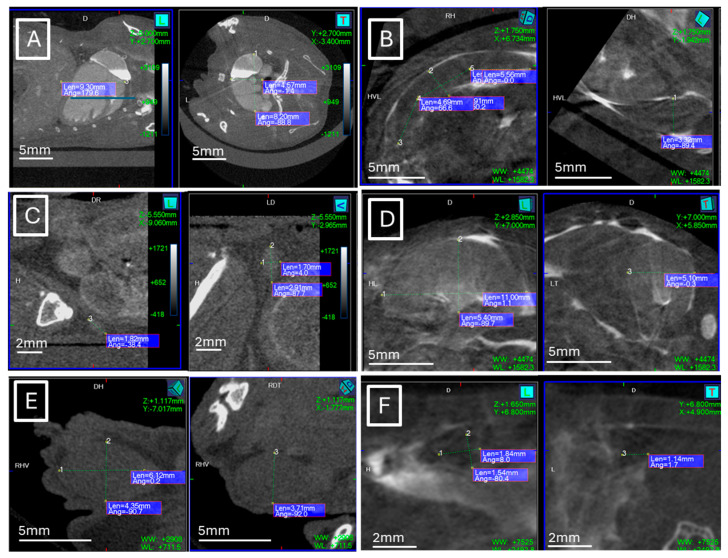
Example of maximal diameter measurements (dotted lines) in microCT studies. (**A**) Heart in coronal (left) and axial (right) views, (**B**) spleen in axial (left) and coronal (right) views, (**C**) left lymph node in sagittal (left) and coronal (right) views, (**D**) left kidney in sagittal (left) and axial (right) views, (**E**) left testicle in sagittal (left) and coronal (right) views, (**F**) left ovarian in sagittal (left) and axial (right) views.

**Figure 3 jimaging-11-00200-f003:**
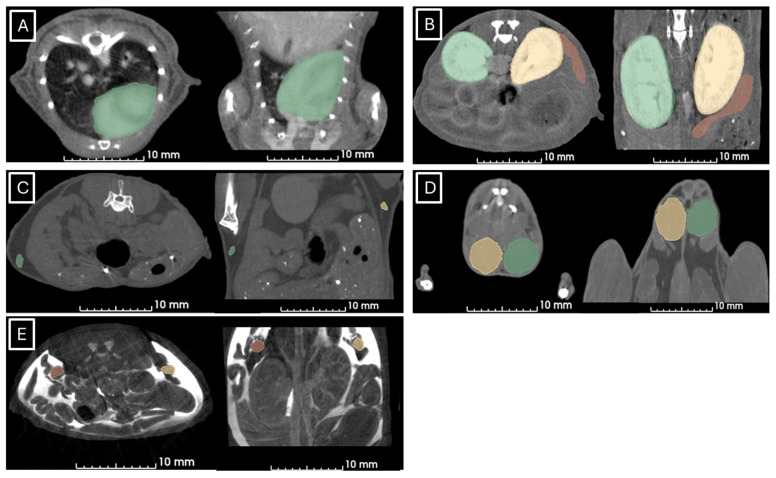
Representative images of manual segmentation in microCT studies. (**A**) Segmented heart in green in axial (left) and coronal (right) views, (**B**) left kidney in yellow, right kidney in green, and spleen in red segmented in axial (left) and coronal (right) views, (**C**) left (in green) and right (in yellow) lymph nodes in axial (left) and coronal (right) views, (**D**) left (yellow) and right (green) testicles in axial (left) and coronal (right) views, (**E**) left (orange) and right (yellow) ovaries in axial (left) and coronal (right) views.

**Figure 4 jimaging-11-00200-f004:**
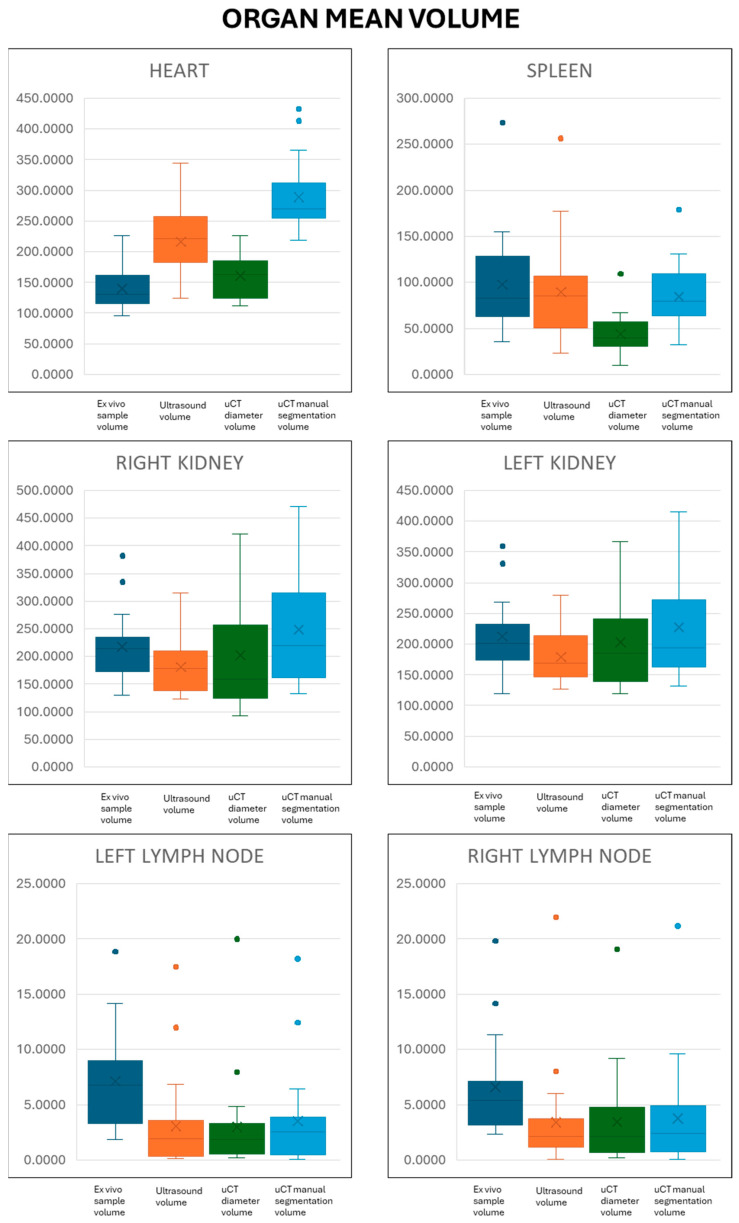
Graphic charts of the mean value for the different organs and volumes. For each organ, from left to right, ex vivo sample volume, ultrasound volume, microCT diameter volume, and microCT manual volume. The box represents the interquartile range, upper and lower lines are maximum and minimum values, isolated points represent outliers, and crosses indicate mean values and inside-the box line median values.

**Figure 5 jimaging-11-00200-f005:**
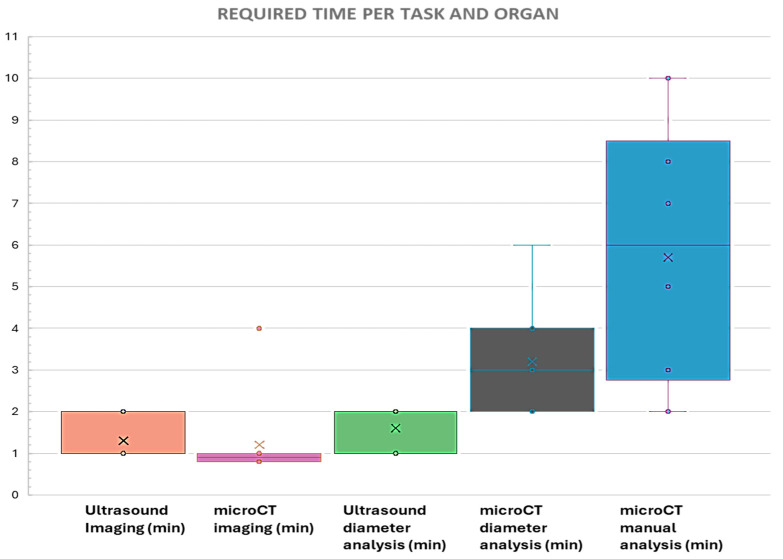
Representation of the time required for each task and organ, including image acquisition and analysis. Note that, in terms of image acquisition, for microCT imaging, some of the organs were scanned simultaneously. The outlier in microCT imaging was the heart, which required 4 min to image. The remaining times required to obtain microCT images of each organ were calculated dividing the total acquisition time by the number of organs scanned.

**Table 1 jimaging-11-00200-t001:** Mean results for each analyzed organ. Weight and specific density are necessary to calculate the sample volume. The other volumes are obtained by image analysis. Weight measured in milligrams, density in mg/mm^3^, volumes in mm^3^.

	Weight (mg)	Density (mg/mm^3^)	Sample Volume (mm^3^)	Ultrasound Volume (mm^3^)	µCT Diameter Method Volume (mm^3^)	uCT Manual Segmentation Method Volume (mm^3^)
**Heart**	147.333	1.053	139.918	216.328	160.967	289.238
**Spleen**	102.567	1.050	97.683	89.366	41.959	84.382
**Right Kidney**	231.789	1.066	217.438	180.937	202.351	248.398
**Left Kidney**	226.711	1.066	212.675	179.179	202.897	227.402
**Right Ovary**	9.283	1.040	8.926	4.950	4.789	5.013
**Left Ovary**	10.217	1.040	9.824	4.462	4.707	5.568
**Right Testicle**	87.767	1.040	84.391	78.500	72.286	70.554
**Left Testicle**	83.817	1.040	80.593	73.387	76.893	76.746
**Right Lymph node**	7.000	1.030	6.796	3.858	3.131	3.768
**Left Lymph node**	7.529	1.030	7.310	3.061	2.736	3.503

**Table 2 jimaging-11-00200-t002:** Summary of *p*-values obtained from correlation analysis between ex vivo sample volumes and volumes obtained by imaging techniques. *p*-values less than 0.05 (underlined) indicate a statistical correlation between the values. In these cases, the volumes obtained by analyzing imaging studies are accurate compared to the gold standard (ex vivo sample volume). For *p* values greater than 0.05, there is no correlation between the volumes, the volumes obtained from the imaging technique and the analysis protocol are not accurate compared to the gold standard.

		Ultrasound Volume	uCT Diameters Method Volume	uCT Manual Segmentation Method Volume
**Ex vivo sample volume**	Heart	0.247	0.0766	0.0433
	Spleen	0.0000262	0.000264	0.0000104
	Left kidney	0.0183	0.0001	0.00123
	Right kidney	0.000051	0.00233	0.0000892
	Left lymph node	0.000005	0.0000186	0.00000881
	Right lymph node	0.000077	0.0018	0.0025
	Left ovary	0.521	0.787	0.491
	Right ovary	0.164	0.335	0.195
	Left testicle	0.0353	0.00361	0.00265
	Right testicle	0.0173	0.0292	0.0705

## Data Availability

The raw data supporting the conclusions of this article will be made available by the authors on request.

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
