# Peer review of "Comparative Analysis of Ultrasonography and MicroCT Imaging for Organ Size Evaluation in Mice"

_2313-433X, 2025, doi:10.3390/jimaging11060200_

Round 1

Reviewer 1 Report

Comments and Suggestions for Authors

See attached file

Reviewer 2 Report

Comments and Suggestions for Authors

Camara Serrano and colleagues reported that their study assessed microCT and ultrasound imaging techniques for measuring organ volumes in mice using ellipsoidal and manual segmentation methods. Both modalities showed good agreement with ex vivo volumes for most organs, except the ovaries and heart. Ovarian volumes were consistently underestimated, and only microCT with manual segmentation provided accurate heart volume measurements. While ultrasound is faster in both acquisition and analysis, it is more operator-dependent, whereas microCT is less dependent during acquisition but requires more time for analysis. Overall, both methods are suitable for preclinical organ volume assessment, with the choice depending on organ shape, equipment availability, and operator experience—favoring the ellipsoidal method for regular organs and manual segmentation for irregular ones. This manuscript meets the standards required for submission to the Journal of Imaging after minor revisions.

  1. What factors contributed to the consistent underestimation of ovarian volume across both imaging techniques, and how might this be addressed in future studies?
  2. Rewrite all the references provided in the manuscript in the Imaging format for clarity and consistency.

Reviewer 3 Report

Comments and Suggestions for Authors

The authors compared two imaging techniques, ultrasound and microCT, for measuring organ volumes in mice, with the idea of expediting such methods for animal studies.  They sought to test the robustness of the imaging modalities by correlating the results with organ volumes calculated from weights and published organ densities.  They also assessed the efficiency and feasibility of the imaging methods for studies by determining the time required to complete them for replicate data.  The formula for calculating organ volume from density on page 4 is incorrect.  It should be volume = weight/density, not weight x density.  In Table 2, there are large differences between CT and US results, with high CVs.  What are the units?  In Table 4, there is no way to evaluate the data.  What do the figures represent?  They are not correlation coefficients.  The description of statistical analyses is inadequate to understand what is being done.  There should be a much more systematic presentation of the data and its evaluation.  Tables of organ weights, published densities and calculated volumes should be juxtaposed with organ volumes measured by the imaging methods, with algorithms--spheres, ellipsoids and integrative methods.  The sentence of lines 84-87 is not clear.  Once the data is presented in a more definitive way, the authors should evaluate the study more critically.

Round 2

Reviewer 1 Report

Comments and Suggestions for Authors

Author Response

Thank you for your review and comments. Please see the attachment for a detailed explanation about the corrections we made in the manuscript.

Reviewer 3 Report

Comments and Suggestions for Authors

The changes in the manuscript should be highlighted or shown in a different colored font.  Tables remaining in the manuscript should be numbered consecutively from 1.  Tables in the supplementary file should be labeled consecutively from S1.  All supplementary tables should be designated as S followed by a number and referred to the same way in the text of the manuscript.  In Table 3 (should be Table 1), weights must be mg and densities mg/mm3.  In Table 6 (should be Table S5), the correlations are still not clear.  Are they ultrasound or CT vs. volumes calculated from weights and densities?  Are the p-values probabilities that the correlations are significantly different from zero?

Comments on the Quality of English Language

The problem is not just grammar or syntax.  The language lacks clarity.  A good example is the statistics section at the end of the Methods.  What is being compared?

Author Response

Thank you for your comments and revisions on the manuscript. Please see the attachment for detailed explanations about the modification we did in our document.

Round 3

Reviewer 1 Report

Comments and Suggestions for Authors

# To the authors

## General remarks

Thanks for implementing major improvements in the manuscript.

I honestly think that this version should have been submitted for peer-review, and not the first two versions!

The presentation of the data is much clearer and paints a better picture of the process of the authors.

## Detailed comments

- Line 28ff: The 3R principles are Replacement, Reduction and Refinement, not reduction, refinement, and repositioning as mentioned here in the text.

- Line 52: micro-SPECT should be microSPECT, according to how the other "micro" modalities are written in the text (microCT, microPET).

- Line 129: "50 um voxel size" should be "50 μm voxel size"

- Lines 168 and 254: "Ex vivo" should be "ex vivo".

- Line 230: Maybe simplify the start of the result section to something like "A total of 184 organ samples were analyzed, derived from 23 animals across 8 target structures. We analyzed each sample with three three imaging analysis protocols, resulting in 552 data points."

- Headers in Table 1: "uCT" should be "μCT"

- Line 245 and others: "Graphic X" should be replaced by "Figure X" throughout the document.

- Line 579: I think that 'election' is the wrong word here, as in my mind this relates to politicians. Maybe change "the election of one or the other" to "the choice between them".

Author Response

Thank you very much for your comments and work during the revision process. We deeply value your time and work with our manuscript, and tried to follow your positive comments and recommendations.

We totally agree, this version should be the one sent for first revision. We really apologize for that,it was our fault due to external pressures and hurrynesses. And thanks again for working so hard into these first versions of the manuscript.

We honestly think we addressed all the minor comments in this version of the manuscript. About the Figure/Graphic point, we tried to differentiate the Figures (coming from the imaging devices) from Graphics (for representing the statistical results and analysis). But of course we are fine putting all of them under the Figure label. Now the document has Figures and Tables.

All the other minor comments are addressed and highlighted in yellow in the manuscript.

Again, thank you very much for your work and time during these revisions. We honestly apprecciate that.

Reviewer 3 Report

Comments and Suggestions for Authors

My concerns have been satisfactorily addressed.  One point: Labeling of supplementary tables is inconsistent.  All the captions to not have the S-designations.  Some of the titles do not have the S-designations.

Author Response

Thank you for your comments and work with our manuscript during the revision process. We totally agree there was an inconsistency within labels in the supplementary data. Everything is now fixed and consistent in the tables titles and footers.
